# Application of an *O*-Linked Glycosylation System in *Yersinia enterocolitica* Serotype O:9 to Generate a New Candidate Vaccine against *Brucella abortus*

**DOI:** 10.3390/microorganisms8030436

**Published:** 2020-03-20

**Authors:** Jing Huang, Chao Pan, Peng Sun, Erling Feng, Jun Wu, Li Zhu, Hengliang Wang

**Affiliations:** State Key Laboratory of Pathogen and Biosecurity, Beijing Institute of Biotechnology, 20 Dongdajie Street, Fengtai, Beijing 100071, China; jinghuangpaper0203@163.com (J.H.); panchaosunny@163.com (C.P.); sunpeng990718@163.com (P.S.); fengel@sohu.com (E.F.); junwu1969@163.com (J.W.)

**Keywords:** *Yersinia enterocolitica* serotype O:9, *Brucella abortus*, *O*-linked glycosylation, bioconjugate vaccine, brucellosis

## Abstract

Brucellosis is a major zoonotic public health threat worldwide, causing veterinary morbidity and major economic losses in endemic regions. However, no efficacious brucellosis vaccine is yet available, and live attenuated vaccines commonly used in animals can cause human infection. *N*- and *O*-linked glycosylation systems have been successfully developed and exploited for the production of successful bioconjugate vaccines. Here, we applied an *O*-linked glycosylation system to a low-pathogenicity bacterium, *Yersinia enterocolitica* serotype O:9 (*Y. enterocolitica* O:9), which has repeating units of O-antigen polysaccharide (OPS) identical to that of *Brucella abortus* (*B. abortus*), to develop a bioconjugate vaccine against *Brucella*. The glycoprotein we produced was recognized by both anti-*B. abortus* and anti-*Y. enterocolitica* O:9 monoclonal antibodies. Three doses of bioconjugate vaccine-elicited *B. abortus* OPS-specific serum IgG in mice, significantly reducing bacterial loads in the spleen following infection with the *B. abortus* hypovirulent smooth strain A19. This candidate vaccine mitigated *B. abortus* infection and prevented severe tissue damage, thereby protecting against lethal challenge with A19. Overall, the results indicated that the bioconjugate vaccine elicited a strong immune response and provided significant protection against brucellosis. The described vaccine preparation strategy is safe and avoids large-scale culture of the highly pathogenic *B. abortus*.

## 1. Introduction

Brucellosis is one of the most common zoonotic diseases worldwide and it has re-emerged in several countries in recent years [1]. More than 500,000 new cases of brucellosis are reported each year [2], and millions of livestock are either infected or at risk [3]. Incidence is likely significantly underestimated because of the wide distribution, mild clinical features, and frequent misdiagnosis of the disease [4]. Brucellosis is mainly caused by *Brucella* spp., which are Gram-negative and facultative intracellular α-proteobacteria. *Brucella melitensis*, *Brucella abortus*, and *Brucella suis* are the most virulent of these species. Humans are mainly infected by direct contact with infected animals or through the consumption of non-pasteurized dairy products [5]. In humans, infection is chronic and patients gradually lose their ability to work. Although rarely fatal, human brucellosis is very serious [6,7], requiring long-term antibiotic treatment and often causing permanent sequelae [8].

At present, animal vaccines against *Brucella* are live attenuated vaccines, and these have played an important role in controlling the spread of brucellosis [9]. However, the efficacy of these vaccines is not ideal [10]—they can cause abortions and infertility in animals, disease in humans [11,12], and do not confer sufficient protection [9,13,14]. For example, S19 and RB51 are two commonly used live attenuated vaccines against *B. abortus* but despite being attenuated, these vaccines can result in infections in humans and are also secreted into the milk of immunized animals [15]. RB51 is resistant to rifampicin and penicillin, making clinical treatment challenging. In addition, evaluation of the live attenuated vaccine often requires the use of virulent smooth strains in biosafety level-3 laboratories, which is not conducive to extensive research. However, there are currently no better alternatives [9,16,17] and, most importantly, there is still no human brucellosis vaccine available [4,18]. Efforts toward further attenuation have ultimately failed [19,20].

The surface polysaccharide of Gram-negative bacteria is an immunodominant antigen, and studies have shown that antibodies against *B. abortus* O-antigen polysaccharide (OPS) elicited by active immunization can provide a protective effect in mice [2,21]. Similarly, Jacques et al. showed that immunization of mice with *B. melitensis* OPS covalently linked to BSA induces protective effects [22]. These results demonstrate that antibodies against OPS are associated with anti-*Brucella* effects and may play an important role in host defense [23]. Therefore, the development of a brucellosis vaccine based on OPS is a promising avenue for the prevention of brucellosis.

Polysaccharide conjugate vaccines produced via the covalent conjugation of bacterial surface polysaccharides to proteins have been very successful, with several efficacious vaccines of this type having been produced using chemical methods and approved for market release [24]. However, this method involves multiple steps and is time-consuming, making it costly and difficult to apply in developing countries and poorer areas [25]. With the discovery of bacterial glycosylation systems and the development of synthetic biology, the preparation of conjugate vaccines using biosynthetic methods has become a hot topic [26,27]. It has been demonstrated that an *N*-linked glycosyltransferase, PglB, from *Campylobacter jejuni* can be expressed in *Escherichia coli* and catalyze the glycosylation of its natural substrate, AcrA [28]. Subsequently, various vaccines based on the PglB glycosylation system were developed, including vaccines against *Shigella dysenteriae* type I [29], *Francisella tularensis* [30], *Staphylococcus aureus* [31], and *Streptococcus pneumoniae* [32,33]. Several of these vaccines have entered clinical trials [25]. Researchers also identified another *O*-linked glycosyltransferase, PglL, from *Neisseria meningitidis* that has broader substrate specificity and, therefore, transfers a greater variety of polysaccharides for bioconjugate vaccine preparation [34]. One such example is the transfer of galactose at the reducing end of the OPS of *Salmonella enterica* serovar Paratyphi A [35]. In recent years, a novel PglS (an *O*-linked oligosaccharyltransferase from *Acinetobacter* spp.) glycosylation system was developed for pneumonia capsule polysaccharide glycosylation [36].

*Yersinia enterocolitica* can be divided into 60 serotypes and six biological types. A highly pathogenic strain is *Y. enterocolitica* bioserotype 1B/O:8, and *Y. enterocolitica* O:9 is a Class II biosafety hazard organism [22,37]. *Y. enterocolitica* O:9 has an identical structure of repeat units in OPS to that of *B. abortus* [37]. Therefore, a bioconjugate vaccine against *B. abortus* could be produced in *Y. enterocolitica* O:9 without biosafety risk. Previously, glycosylated AcrA was obtained in *Y. enterocolitica* O:9 with the *N*-linked glycosylation system, although this glycoprotein generated an IgG antibody response but failed to exert a protective effect in mice [22].

In this study, we produced a bioconjugate vaccine against *B. abortus* by introducing an *O*-linked glycosylation system into *Y. enterocolitica* O:9, which avoided the cloning of large fragments of bacterial polysaccharide-encoding genes and the low efficiency of exogenous expression. Besides its biosafety advantages, *Y. enterocolitica* O:9 is easy to manipulate and culture, while *B. abortus* grows slowly and requires specific culture conditions. Through the catalysis of PglL, the OPS from *Y. enterocolitica* O:9 could be conjugated to the recombinant cholera toxin B subunit (rCTB). A series of animal experiments showed that our bioconjugate vaccine was safe and could induce effective immune responses in mice, providing a potential candidate vaccine for the prevention and control of brucellosis. Moreover, the evaluation of our vaccine made use of the hypovirulent smooth strain of *B. abortus,* which is less pathogenic than the virulent smooth strain.

## 2. Materials and Methods

### 2.1. Bacterial Strains, Growth Conditions, and Plasmids

*Y. enterocolitica* O:9 strain 52212 (YeO9_52212) and the *B. abortus* hypovirulent smooth strain A19 were provided by the Institute for Communicable Disease Control and Prevention, Chinese Centre for Disease Control and Prevention, Beijing, China. YeO9_52212 was cultured in Brain Heart Infusion (BHI) medium at 25 °C. To induce plasmid expression in the bacteria, cells were cultured to an optical density at 600 nm of 0.6–0.8, followed by the addition of isopropyl-β-D-thiogalactopyranoside (IPTG) to a final concentration of 1 mM and further culture of the cells at 25 °C for 14 h. *B. abortus* strain A19 was cultured in Tryptic Soy Broth (TSB) medium at 37 °C. Solid media contained 1.5% agar. Plasmids (pET-pglL-CTB4573H and pET-CTB4573H) were constructed based on our previous work [38]. Kanamycin (50 μg/mL) was used for plasmid selection as required.

### 2.2. Western Blotting

IPTG-induced cells (1 mL) were collected by centrifugation and then resuspended in 100 µL of ddH_2_O. The whole-cell or purified samples were mixed with 1:1 (v/v) 1× Loading Buffer (100 mM Tris-HCl, pH 6.8, 3.2% (w/v) SDS, 0.04% (w/v) bromophenol blue, 16% (v/v) glycerol, and 40 mM DL-dithiothreitol). The samples were placed in a boiling water bath for 10 min and then cooled to room temperature. After centrifugation at 10,000 ×*g* for 10 min, 10 μL of each sample was subjected to SDS-PAGE and western blotting using the methods described in our previous work [38]. Horseradish peroxidase-conjugated 6×His-tag antibody (Abmart, Shanghai, China) was used to detect the His-tagged proteins. A *B. abortus* monoclonal antibody (Thermo Fisher Scientific, Waltham, MA, USA) was diluted 1:100 and used to detect polysaccharides of *B. abortus* in the glycoproteins. A *Y. enterocolitica* O:9 monoclonal antibody (Fitzgerald, Acton, MA, USA) was diluted 1:200 and used to detect *Y. enterocolitica* O:9 polysaccharide in the glycoproteins.

### 2.3. Glycoprotein Purification

IPTG-induced cells (7 L) were collected by centrifugation and then resuspended in 300 mL of Buffer A1 (20 mM Tris-HCl, pH 7.5, 10 mM imidazole, and 0.5 M NaCl), after which the cells were disrupted by ultrasound for 2 h. The supernatant of the cell lysate was harvested by centrifugation at 10,000 ×*g* for 20 min and then loaded onto a nickel affinity column (cOmplete His-Tag Purification Resin, Roche, penzberg, Germany) that had been pre-equilibrated with five column volumes of Buffer A1. The flow rate was 4 mL/min. After washing with ten column volumes of Buffer A1, the target glycoprotein was eluted in Buffer A2 (20 mM Tris-HCl, pH 7.5, 0.5 M imidazole, and 0.5 M NaCl). The eluent was concentrated with a 10 kDa cut-off centrifugal filter (Merck, Darmstadt, Germany) to less than 10 mL. The concentrated solution was separated through a Superdex 200 Prep Grade column (16 mm × 1000 mm; GE Healthcare, Beijing, China) with phosphate-buffered saline (PBS) at a flow rate of 1 mL/min. The fractions were collected and analyzed by 12% SDS-PAGE.

### 2.4. Lipopolysaccharide (LPS) and OPS Extraction

LPS extraction was performed as described previously [35]. Briefly, the culture was collected by centrifugation and then resuspended in ddH_2_O. After freezing and thawing, an equal volume of 90% phenol was added to each sample followed by vigorous shaking at 68 °C and then centrifugation to collect the uppermost part of the supernatant (the water layer). The phenol layer was re-extracted with ddH_2_O and the previous step was repeated. The extracts were dialyzed into ddH_2_O, then DNase (5 µg/mL; Solarbio, Beijing, China), RNase (1 μg/mL; Solarbio), and proteinase K (20 μg/mL; Merck) were sequentially added to the dialyzed samples and incubated at the optimal temperature for each enzyme. After a boiling water bath for 10 min, samples were centrifuged to obtain the LPS.

Glacial acetic acid was added to the extracted LPS to achieve a final concentration of 1% (v/v). After boiling for 90 min, the pH of the LPS extract was adjusted to 7.0 with NaOH. Finally, the mixture was centrifuged at 40,000 ×*g* for 5 h, and the supernatant was collected as the OPS fraction.

### 2.5. Animal Immunization Experiments

Specific-pathogen-free female BALB/c mice were purchased from Charles River and housed in the Laboratory Animal Centre of the Academy of Military Medical Sciences at constant temperature and humidity. Water, food, and bedding were changed every 4 days. All animal experiments were approved by and conducted in accordance with the recommendations of the Animal Care and Use Committee of the Academy of Military Medical Sciences (ethics approval code IACUC-DWZX-2017-023, approved in November 2017).

Groups of 10 5-week-old BALB/c mice were used in the immunization experiments. Purified glycoproteins or OPS were diluted with PBS, and aluminum hydroxide adjuvant (General Chemical Corp, Brighton, MI, USA) was added to 10% of the final volume. Mice were immunized intraperitoneally (i.p.) with three doses (2.5 μg of the polysaccharides/mouse) at 2-week intervals and blood samples were collected by tail snip on the 10th day following the third dose. Mouse sera were separated from the blood and stored at −20 °C. Two weeks after the final immunization, mice were injected i.p. with different doses of A19 as described below, followed by determination of the bacterial load in the mouse spleens and monitoring of mouse survival.

### 2.6. Enzyme-Linked Immunosorbent Assay (ELISA)

A 96-well immunoplate was precoated with 10 μg/mL poly-L-lysine (100 μL per well). ELISA was performed as described previously [35]. Briefly, the 96-well immunoplate was coated for 2 h with diluted LPS at 37 °C and then washed three times with Wash Buffer (PBS + 0.05% Tween 20). The plates were patted dry and Blocking Buffer (PBS + 5% milk powder) was added to each well followed by incubation at 37 °C for 2 h. After drying the plates, diluted serum from the immunized mice was added to each well and the plate was incubated at 37 °C for 1 h. After another washing and drying step, 1:15,000 diluted HRP-conjugated goat anti-mouse IgG antibody (Abcam, Shanghai, China) was added to each well and incubated at 37 °C for 1 h. The washing and drying step was repeated. The Soluble TMB Kit (CWbio, Beijing, China) was used to initiate the detection reaction. Stop solution (2 M H_2_SO_4_) was added to each well to stop the reaction, and the absorbance of each well was measured at a wavelength of 490 nm with a microplate reader.

### 2.7. Determination of Bacterial Loads in Spleens

Mice were immunized and a non-lethal dose of *B. abortus* A19 was injected i.p. on the 14th day after the third immunization. On the 7th day post-infection, mice were sacrificed by neck dissection, and the spleens were removed and placed in 1.5 mL microcentrifuge tubes. The spleens were weighed and then homogenized with normal saline. After centrifugation at 5000 ×*g* for 10 min, the supernatant was discarded and the pellet was resuspended with normal saline. After repeating the previous step twice, the pellet was resuspended with 1 mL of 0.1% (w/v) sodium deoxycholate. Finally, the bacterial suspension was diluted with normal saline and cultured on solid TSB medium. Bacterial colonies were counted after 2 days of culture at 37 °C.

### 2.8. Determination of Cytokine Levels

The cytokine levels of the immunized mice were determined with the Mouse TNF-α Precoated ELISA Kit, Mouse IL-1β Precoated ELISA Kit, Mouse IL-2 Precoated ELISA Kit, and Mouse IL-6 Precoated ELISA Kit (Dakewe, Shenzhen, China) in accordance with the manufacturer’s instructions. Briefly, diluted serum samples and standards were added to the pre-coated wells, followed by the addition to each well of the biotinylated antibody relevant to the kit being used and incubation at 37 °C for 90 min. The wells were washed four times with Washing Buffer and patted dry, after which, Streptavidin-HRP was added to each well and the plates were incubated at 37 °C for 30 min. After another washing and drying step, TMB was added to each well and incubated at 37 °C for 15 min shielded from the light, and the reactions were terminated using Stop solution. The absorbance of each well was measured at a wavelength of 450 nm with a microplate reader.

### 2.9. Hematoxylin and Eosin (HE) Staining

The livers and spleens of immunized mice were fixed with 4% paraformaldehyde (Solarbio, Beijing, China), and then paraffin-sectioned and stained using the Hematoxylin-Eosin (HE) Staining Kit (Solarbio) in accordance with the manufacturer’s instructions. Briefly, the fixed samples were embedded and sectioned, then conventional dewaxing and hydration were performed. The tissue sections were stained with HE, after which, dehydration, cleaning for transparency, and neutral resin sealing were performed successively.

### 2.10. Statistical Analysis

Antibody titers were log_2_-transformed and bacterial loads were log_10_-transformed. Statistical analyses were conducted using GraphPad Prism version 7.0 (GraphPad, San Diego, CA, USA).

## 3. Results

### 3.1. Application of the O-linked Glycosylation System in Y. Enterocolitica O:9

Studies have reported that most polysaccharides are recognized by PglL [34], and our previous works demonstrated the successful use of PglL in some Gram-negative bacteria [35,38]. To create the conjugate vaccine against *B. abortus* with the *O*-linked glycosylation system, we first needed to verify that the OPS of *B. abortus* could be recognized by PglL. *Y. enterocolitica* O:9 strain 52212 (YeO9_52212) was used as the host cell because this low-pathogenicity strain has an identical OPS structure to that of *B. abortus* [37]. The pET-pglL-CTB4573H plasmid, from which PglL and rCTB [35] were co-expressed, or the pET-CTB4573H plasmid, from which rCTB alone was expressed, were introduced into YeO9_52212. After induction with IPTG and overnight culture, the total protein was extracted and separated by SDS-PAGE. Coomassie Blue staining and western blotting showed that the molecular weight (MW) of rCTB when expressed alone was only around 15 kDa (Figure 1A). The MW of rCTB increased when PglL was co-expressed, indicating that rCTB might have been glycosylated. Further, two bands for rCTB were observed in YeO9_52212 cells expressing pET-pglL-CTB4573H. The lower band showed a slightly shifted MW compared with CTB4573H and the higher band had a MW between 35–40 kDa. These results indicated that the *O*-linked glycosylation system was efficiently expressed in YeO9_52212 and that almost all of the substrate protein had been glycosylated.

Glycoproteins were isolated by Ni^2+^ affinity and size exclusion chromatography (Appendix A). The recovery efficiency following purification was approximately 150 µg polysaccharide/L medium. Anti-*B. abortus* and anti-*Y. enterocolitica* O:9 specific monoclonal antibodies were used to test the structural similarity of OPS between *Y. enterocolitica* O:9 and *B. abortus*. The higher MW band of glycosylated CTB4573H (CTB-OPS_Ba_) was detected by both antibodies, while the lower band was not observed because of the poor immunogenicity of short-chain OPS (Figure 1B). CTB naturally exists in a pentameric form [39]. To verify whether glycosylated CTB4573H was also in a polymeric state, we performed native gel electrophoresis. Coomassie Blue staining showed that the MW of CTB-OPS_Ba_ (C-OPS_Ba_) was around 242 kDa (Figure 1C), which is approximately 5-fold that of the C-OPS_Ba_ monomer (~40 kDa). Thus, rCTB could still pentamerize after glycosylation and purification.

### 3.2. Induction of Specific Antibody Responses in Mice

After confirming the conservation of C-OPS_Ba_ between *Y. enterocolitica* O:9 and *B. abortus*, we next assessed the immunogenicity of C-OPS_Ba_ through a series of animal experiments. After a preliminary assessment of the safety of this vaccine obtained by measuring biochemical indicators including ALP, AST, ALT, and BUN (Appendix A), we tested the serum titers of antibodies against YeO9_52212 LPS following C-OPS_Ba_ immunization.

Ten 5-week-old BALB/c mice were immunized i.p. with either purified C-OPS_Ba_ or C-OPS_Ba_ adjuvanted with aluminum hydroxide (C-OPS_Ba_+Al) on Days 0, 14, and 28. Each group was immunized with the same quantity of polysaccharide (2.5 µg/mouse) and another group was immunized with PBS only as a control. On the 10th day following the third immunization, serum was collected by tail snip and used to characterize IgG responses against YeO9_52212 LPS by ELISA. C-OPS_Ba_ was observed to induce antibody titers against YeO9_52212 LPS in mice (Appendix A). We further detected specific antibodies against *B. abortus* LPS.

Purified C-OPS_Ba_ and OPS_Ba_ from YeO9_52212, with or without 10% aluminum hydroxide adjuvant (C-OPS_Ba_, C-OPS_Ba_+Al, OPS_Ba_, OPS_Ba_+Al), were used for immunization as described above. ELISA results showed that *B. abortus* LPS-specific IgG titers in all groups were elevated compared with the PBS-vaccinated group, especially in C-OPS_Ba_- and C-OPS_Ba_+Al-vaccinated mice. The titer in the C-OPS_Ba_ group was significantly higher than the titers of the OPS_Ba_ and OPS_Ba_+Al groups but similar to that of the C-OPS_Ba_ +Al group (Figure 2A). We further detected IgG subclass titers (IgG1, IgG2a, IgG2b, and IgG3) against *B. abortus* A19 LPS in the sera of PBS-, OPS_Ba_-, and C-OPS_Ba_-vaccinated mice. Apart from the IgG3 titers, other subclass titers in the C-OPS_Ba_ group were significantly higher than in the PBS group (Figure 2B).

### 3.3. Evaluation of Vaccine-Induced Protection in Mice following Infection with a Non-Lethal dose of B. Abortus

Given the success of the bioconjugate vaccine in eliciting both *Y. enterocolitica* O:9 and *B. abortus* LPS-specific IgG antibodies, we further evaluated the protective effects of this vaccine against non-lethal infection. For traditional live attenuated vaccines against *B. abortus*, the virulent smooth strain of *B. abortus* has to be used to evaluate the protective effects (mainly through measurement of the clearance rate of bacteria in the mouse spleen). In contrast, there is no unified evaluation standard for the bioconjugate type of vaccines, and preliminary evaluation of their protective effect can be evaluated through infection with the *B. abortus* hypovirulent smooth strain. Fourteen days after the third immunization, mice were challenged with a non-lethal dose (1.03 × 10^7^ CFU/mouse) of *B. abortus* hypovirulent strain A19 (A19). For all treatment groups, the TNF-α levels began to decline after reaching a peak on the 5th day (Figure 3A). Serum TNF-α levels in the PBS-immunized group and the OPS_Ba_- and OPS_Ba_+Al-immunized groups increased markedly following infection. In contrast, C-OPS_Ba_ and C-OPS_Ba_+Al vaccination significantly inhibited the production of TNF-α, and TNF-α levels in the C-OPS_Ba_ group were lower than those in the C-OPS_Ba_+Al group. We also measured serum IL-1β, IL-2, and IL-6 levels in the immunized mice but no significant differences in expression were detected.

Seven days following infection, the mice were dissected and their spleens were removed. We found that the spleen size had increased following infection, with a significant increase in the spleen weight in the PBS group compared with the C-OPS_Ba_ group, and in the OPS_Ba_ group compared with the control group (Appendix A and Figure 3B). We enumerated the bacteria in each spleen and observed approximately 10^3^–10^3.5^ CFU of A19 in C-OPS_Ba_ and C-OPS_Ba_+Al-vaccinated mice. In contrast, the bacterial loads of other groups remained around 10^4^–10^4.5^ CFU in the spleen, which was significantly higher than those of mice immunized with C-OPS_Ba_ and C-OPS_Ba_+Al (Figure 3B). These results indicated that mice immunized with C-OPS_Ba_ and C-OPS_Ba_+Al were effectively protected against invading pathogens and the resulting lethal cytokine storms.

Spleens and livers were paraffin-sectioned and stained with HE. Compared with the control mice, the immunized mice showed different degrees of pathological changes in their spleens and livers. The spleens of all vaccinated mice with low-spectroscopy observation showed an increase in the white pulp, reflecting the proliferation of lymphocytes. Multinucleated giant cells with varying numbers and shapes of nuclei caused by foreign bodies increased with high-spectroscopy observation (green arrows) (Figure 3C). The livers showed an aggregation of lymphocytes under low-spectroscopy observation and hyperplastic nodules caused by *Brucella* were seen under high-spectroscopy observation (blue arrows) (Figure 3C). Spleen and liver damage were most severe in mice immunized with PBS, followed by OPS_Ba_ and OPS_Ba_+Al. This damage was mildest in the C-OPS_Ba_ and C-OPS_Ba_+Al groups, especially the C-OPS_Ba_ group (Figure 3C).

### 3.4. Evaluation of Protection against the Lethal dose Challenge

After demonstrating the safety and efficacy of the vaccine against non-lethal infection, we next evaluated its protective efficacy against lethal challenge. Separate groups of mice were immunized with PBS, OPS_Ba_, OPS_Ba_+Al, C-OPS_Ba_, or C-OPS_Ba_+Al as described previously and challenged with a lethal dose of A19 (Figure 4A). Mice were challenged i.p. with approximately 3 × LD_50_ of A19 (1.54 × 10^8^ CFU/mouse) 2 weeks after the last vaccination, and survival was monitored for up to 14 days. All of the PBS-vaccinated mice died within 2 days and the 2-week survival rates in the OPS_Ba_ and OPS_Ba_+Al groups were no more than 60%. In contrast, vaccination with C-OPS_Ba_ or C-OPS _Ba_+Al provided 100% protection without the adjuvant (Figure 4B). These results demonstrated that the administration of C-OPS_Ba_ was far more effective than OPS_Ba_ in protecting mice against lethal challenge with *B. abortus*. These findings were consistent with previous studies.

To verify the stability of the protective effect of C-OPS_Ba_ in mice, we immunized another three groups of mice with PBS, C-OPS_Ba_ or C-OPS_Ba_+Al as described above to assess their survival rate following a higher lethal dose challenge (2.51 × 10^8^ CFU/mouse, 5 × LD_50_). With this higher challenge dose, the survival rate of the mice was 90% for the C-OPS_Ba_- and C-OPS_Ba_+Al-vaccinated mice (Figure 4C), indicating that C-OPS_Ba_ can induce potent protection in mice. The IgG titers against A19 LPS in the sera of these mice were then measured. There was a significant difference in titer between the C-OPS_Ba_- and PBS-vaccinated mice (Appendix A). Finally, immunized mice were infected with an even higher dose of A19 (lethal to 20% of PBS-vaccinated mice). The C-OPS_Ba_- and C-OPS_Ba_+Al-vaccinated mice showed significantly lower bacterial loads in the spleen compared with PBS-vaccinated mice (Appendix A). These results indicated that C-OPS_Ba_ can induce stable protection in mice.

## 4. Discussion

In this study, we developed a novel bioconjugate vaccine candidate against *B. abortus* based on a bacterial *O*-linked glycosylation system. YeO9_52212 was engineered to act as the host organism for the production of the bioconjugate vaccine against *B. abortus*. Because YeO9_52212 is less pathogenic and more easily cultured than *B. abortus*, we were able to avoid the large-scale culture of highly pathogenic bacteria. Our results demonstrated that C-OPS_Ba_ induced strong immune responses against *B. abortus* even in the absence of the aluminum hydroxide adjuvant.

We chose OPS as the target for vaccine development because of its ability to elicit antibodies that can confer protection against infection [40]. The glycoprotein generated in YeO9_52212 was able to induce antibodies against *B. abortus* as a result of the similarity in OPS between these two bacterial species [37]. The OPS of *Brucella* spp. is a homopolymer of *N*-formyl-perosamine (4-formamido-4,6-dideoxy D-mannose) with various proportions of α-(1→2) and α-(1→3)-linkages and it has three antibody reactivities (A, M, and C). The A epitope corresponds to five or more continuous sugar units with α-(1→2) linkages and is typical of some *B. abortus* and *B. suis* biovars. The M epitope corresponds to the pentasaccharide with four α-(1→2)-linked and one α-(1→3)-linked polymers of the same sugar and is characteristic of *B. melitensis.* The C epitope represents the ratio of A/M reactivities highly repeated in the OPS of all biovars [2,23]. *B. abortus* contains ~98% of the A epitope, *B. suis* has a unique 1:7 ratio of α-(1→3)- to α-(1→2)-linked polymers, and *B. melitensis* has only the M epitope of the pentasaccharide repeating unit [2]. *Y. enterocolitica* O:9 generates strong cross-reactivity with *Brucella* as its OPS is a homopolymer of *N*-formyl-perosamine with exclusively α-(1→2)-linkages [2,22]. It is thus reasonable to speculate that the bioconjugate vaccine produced in YeO9_52212 would induce cross-reactive immune responses against *B. suis* and *B. melitensis*. Cross-protection against *B. suis* or *B. melitensis* remains to be verified, and this could further expand the scope of vaccine application.

Although the use of AcrA as a carrier protein failed to exert a protective effect in mice against *B. abortus* infection in a previous report, the feasibility of the use of carriers proteins as part of the *B. abortus* vaccine preparation strategy was demonstrated [22]. Since carrier proteins play an important role in immune response, protein toxins such as the tetanus toxoid, the recombinant exotoxin A protein from *Pseudomonas aeruginosa* (rEPA), the diphtheria toxoid, and rCTB used in the present study, are usually included in the vaccine design for better stimulation of the immune system [24]. Our previous studies found that different carrier proteins have different immune effects, and we demonstrated that the use of rCTB in bioconjugate vaccines conferred superior protection against *Shigella flexneri* and *Salmonella enterica* serovar Paratyphi A than rEPA, another commonly used carrier protein [35,38]. The bioconjugate vaccine in the present study used rCTB and exhibited a potent immune effect that was probably attributable to the good immunoadjuvant properties of CTB itself [39], as CTB has been shown to induce anti-inflammatory responses and regulate T-cell immune responses [41]. In addition, the rCTB pentamer can form more complex spatial structures, which facilitates activation of the immune response. We found that protein glycosylation did not change the pentameric structure of the substrate rCTB protein, as shown by the native PAGE results in this study. This finding also suggested that the size, structure, charge, and other physical/chemical properties of carrier proteins should be considered for the rational design of vaccines in the future.

Because there is no clear and unified evaluation model for *Brucella* vaccines, we used TNF-α as an indicator of the protective effect of the vaccine. TNF-α is one of the most important proinflammatory cytokines, and its secretion is associated with inflammatory responses caused by infection. TNF-α is one of the first cytokines released by macrophages following *B. abortus* infection, and its production results from direct interactions between *Brucella* and macrophages [42]. TNF-α plays a crucial role in enhancing IL-12 production [23], and depletion of IL-12 leads to decreased production of interferon-γ and nitric oxide, resulting in exacerbation of infection [23,42]. Here, the serum TNF-α levels in mice immunized with C-OPS_Ba_ were significantly lower than those in the other treatment groups after infection with *B. abortus*. Infection via the intraperitoneal route can induce a strong inflammatory response. There was a higher level of recall response in the C-OPS_Ba_-vaccinated group because of pre-existing protective antibodies, which could quickly neutralize and kill the invading pathogens, such that the bacterial loads in the spleen were significantly reduced. Because the pathogen could be rapidly cleared, there was no strong immune response; in contrast, the mouse spleens were enlarged in the PBS- and OPS_Ba_-immunized groups, with a significantly increased proportion of white pulp observed.

The two most common routes of human infection for Brucellosis are the ingestion of contaminated dairy products or inhalation of contaminated aerosols. This leads to *Brucella* colonization of organs such as the spleen, liver, and lymph nodes. Owing to its potential spread by aerosols, *Brucella* has been classified as a biological threat [43]. This poses a risk during the production of live attenuated vaccines, requiring stricter production requirements and resulting in higher production costs. Thus, the live attenuated vaccines could lead to the possibility of leakage contamination. Research and development of new candidate vaccines are urgently required. Our study provides a conceptual advance for vaccine production against *B. abortus*—not only is the described production process safer and more economical than that used for traditional live attenuated vaccines, but our strategy also avoids the potential risk of virulence reversion and antibiotic resistance. Our conjugate vaccine could also be used to diagnose brucellosis [22]. In addition, subunit vaccines make it easy to distinguish between immunization and natural infection in animals. To further improve safety, in future studies we will delete the virulence-related genes of the YeO9_52212 engineered bacteria to render it even more suitable for industrial production.

## Figures and Tables

**Figure 1 microorganisms-08-00436-f001:**
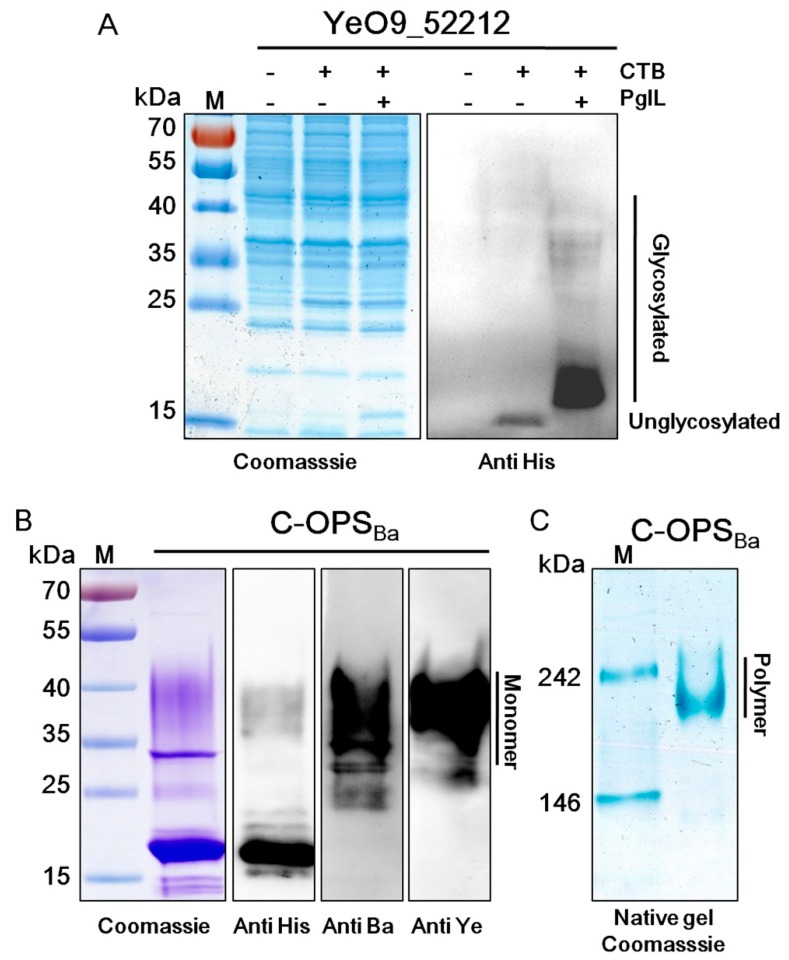
Analysis of glycosylated recombinant cholera toxin B subunit (rCTB) and CTB-OPS_Ba_ expressed in YeO9_52212. (**A**) YeO9_52212 was transformed with pET-CTB4573H or pET-pglL-CTB4573H and then induced with IPTG. A wild type strain was treated in the same way (negative control). Coomassie Blue staining (left) and western blotting (right) were performed. (**B**) The CTB-OPS_Ba_ (C-OPS_Ba_) glycoprotein was purified from strain YeO9_52212 co-expressing PglL and rCTB. Samples were separated by 12% SDS-PAGE and analyzed by Coomassie Blue staining or western blotting using anti-His, anti-*B. abortus* (anti-Ba) or anti-*Y. enterocolitica* O:9 (anti-Ye) antibodies. (**C**) Coomassie Blue staining after native gel electrophoresis of C-OPS_Ba_.

**Figure 2 microorganisms-08-00436-f002:**
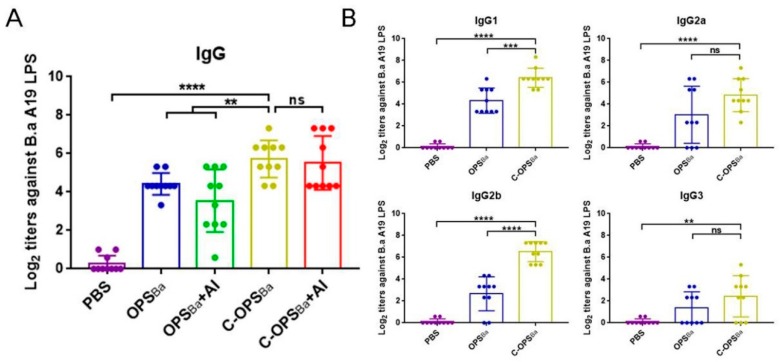
IgG responses against *B. abortus* A19 lipopolysaccharide (LPS). (**A**) IgG titers against A19 LPS were measured in sera from PBS-, OPS_Ba_-, OPS_Ba_+Al-, C-OPS_Ba_-, and C-OPS_Ba_+Al-vaccinated mice. (**B**) IgG subclass titers (IgG1, IgG2a, IgG2b, and IgG3) against A19 LPS were measured in sera from PBS-, OPS_Ba_-, and C-OPS_Ba_-vaccinated mice. Each value represents the mean ± standard deviation of log_2_-transformed titers in the sera of individual mice (shown as data points on the graphs) from each group (*n* = 10). The unpaired *t*-test was used to evaluate differences between IgG titers (**, *p* <0.01; ****, *p* <0.0001; ns, no statistically significant difference).

**Figure 3 microorganisms-08-00436-f003:**
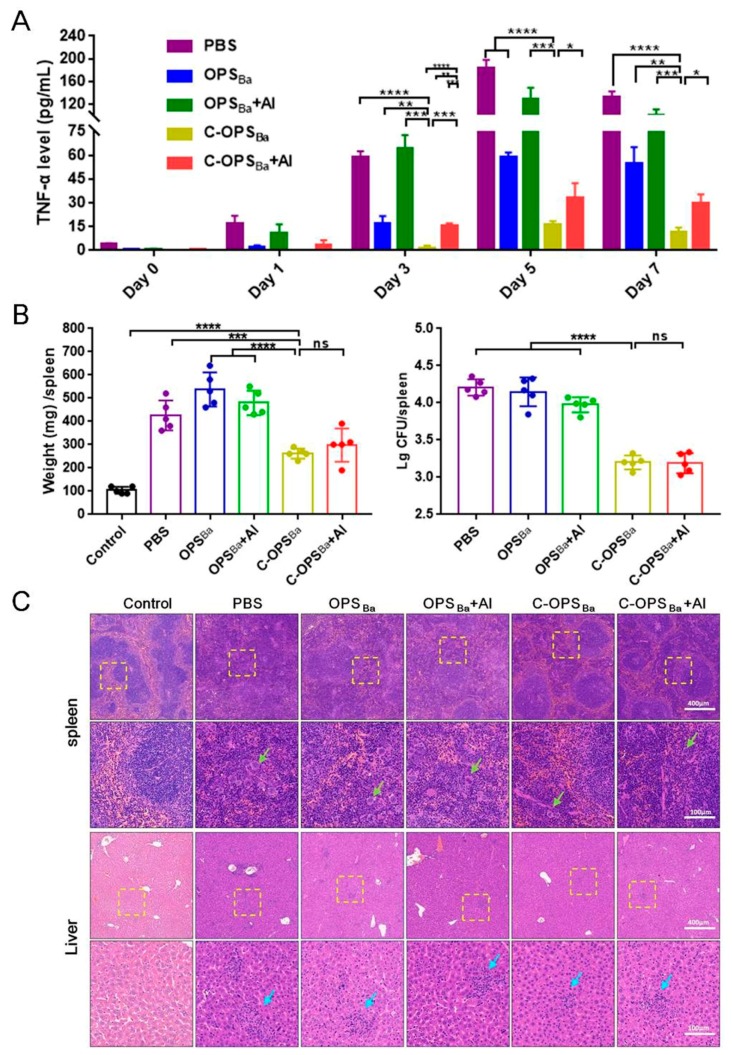
Immune responses of mice following non-lethal *B. abortus* A19 infection. Immunized mice were infected intraperitoneally with 1.03 × 10^7^ CFU of A19 on the 14th day following the third immunization. As a control, another group of naive mice was injected intraperitoneally with normal saline. (**A**) After infection, the sera of mice in each group were collected on the 1st, 3rd, 5th, and 7th day and the TNF-α levels were measured. The unpaired *t*-test was used to evaluate differences between TNF-α levels at different time points. Each value represents the mean ± standard deviation (*n* = 3). (**B**) On the 7th day post-infection, mouse spleens were collected and weighed and the bacterial loads were measured. Each value represents the mean ± standard deviation of spleen weight or log_10_-transformed bacterial loads (CFU/spleen) of individual mice (shown as data points on the graphs) from each group (*n* = 5 per group). The unpaired *t*-test was used to evaluate differences between spleen weights or bacterial loads (***, *p* <0.001; ****, *p* <0.0001; ns, no statistically significant difference). (**C**) The livers and spleens of infected mice and normal mice (Control) were fixed with 4% paraformaldehyde, paraffin sectioned, and then stained with hematoxylin and eosin. The yellow boxes in the top panels represent the field of view in the corresponding figures below, which were magnified four times. Green arrows indicate multinucleated giant cells and blue arrows indicate hyperplastic nodules.

**Figure 4 microorganisms-08-00436-f004:**
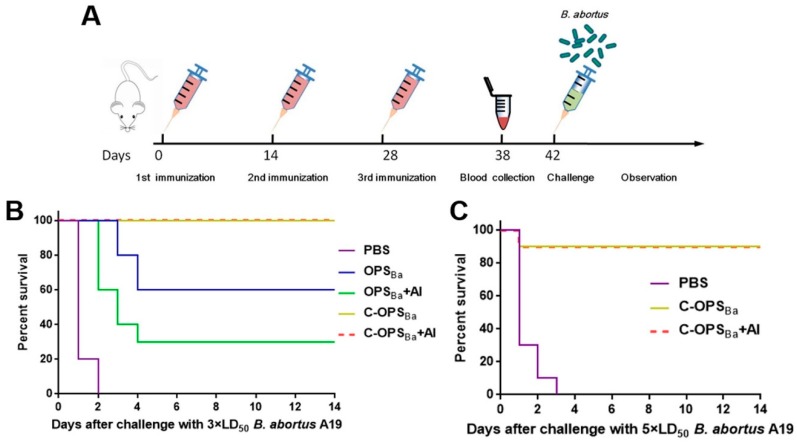
Survival of BALB/c mice after challenge with a lethal dose of *B. abortus* A19. Mice were challenged 2 weeks after final immunization by intraperitoneal injection with different doses of A19 and survival was monitored. (**A**) Schematic diagram of the challenge experiment. (**B**) Mice were immunized with PBS, OPS_Ba_, OPS_Ba_+Al, C-OPS_Ba_ or C-OPS_Ba_+Al and challenged with ~1.54 × 10^8^ CFU/mouse (3 × LD_50_) of A19 (*n* = 10). (**C**) Mice were immunized with C-OPS_Ba_, C-OPS_Ba_+Al or PBS then challenged with approximately 2.51 × 10^8^ CFU/mouse (5 × LD_50_) of A19 (*n* = 10).

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
