# Peer review of "Application of an O-Linked Glycosylation System in Yersinia enterocolitica Serotype O:9 to Generate a New Candidate Vaccine against Brucella abortus"

_microorganisms, 2020, doi:10.3390/microorganisms8030436_

Round 1

Reviewer 1 Report

The current study by Huang et al focuses on the use of a O-linked glycosylation for the production of a novel vaccine against Brucella abortus. The data are well presented and the manuscript is well written. I only have one comment before I recommend the paper for publication.

1) The authors are asked to add more details to their Materials & Methods sections so that potential readers are capable of reproducing the data.

Author Response

Response to Reviewer 1 Comments

Point 1: The authors are asked to add more details to their Materials & Methods sections so that potential readers are capable of reproducing the data.

Response 1: Thank you for this suggestion, we agree with the reviewer. To make our results more repeatable, we have added further detail to the following sections of the Materials and Methods in our revised manuscript:

  1. Section 2.2 (Western blotting), lines 108 to 112

IPTG-induced cells (1 mL) were collected by centrifugation and then resuspended in 100 µL of ddH2O. The whole-cell or purified samples were mixed with 1:1 (v/v) 1× Loading Buffer (100 mM Tris-HCl, pH 6.8, 3.2% (w/v) SDS, 0.04% (w/v) bromophenol blue, 16% (v/v) glycerol, and 40 mM DL-dithiothreitol). The samples were placed in a boiling water bath for 10 min and then cooled to room temperature.

  1. Section 3 (Glycoprotein purification), lines 120 to 126

IPTG-induced cells (7 L) were collected by centrifugation and then resuspended in 300 mL of Buffer A1 (20 mM Tris-HCl, pH 7.5, 10 mM imidazole, and 0.5 M NaCl), after which the cells were disrupted by ultrasound for 2 h. The supernatant of the cell lysate was harvested by centrifugation at 10,000 ×g for 20 min and then loaded onto a nickel affinity column (cOmplete His-Tag Purification Resin, Roche, Germany) that had been pre-equilibrated with five column volumes of Buffer A1. The flow rate was 4 mL/min. After washing with ten column volumes of Buffer A1, the target glycoprotein was eluted in Buffer A2 (20 mM Tris-HCl, pH 7.5, 0.5 M imidazole, and 0.5 M NaCl).

  1. Section 8 (Determination of cytokine levels), lines 180 to 189

The cytokine levels of the immunized mice were determined with the Mouse TNF-α Precoated ELISA Kit, Mouse IL-1β Precoated ELISA Kit, Mouse IL-2 Precoated ELISA Kit, and Mouse IL-6 Precoated ELISA Kit (Dakewe, Shenzhen, China) in accordance with the manufacturer’s instructions. Briefly, diluted serum samples and standards were added to the pre-coated wells, followed by the addition to each well of the biotinylated antibody relevant to the kit being used and incubation at 37 °C for 90 min. The wells were washed four times with Washing Buffer and patted dry, after which, Streptavidin-HRP was added to each well and the plates were incubated at 37 °C for 30 min. After another washing and drying step, TMB was added to each well and incubated at 37 °C for 15 min shielded from the light, and the reactions were terminated using Stop solution. The absorbance of each well was measured at a wavelength of 450 nm with a microplate reader.

  1. Section 9 (Hematoxylin and eosin (HE) staining), lines 191 to 196

The livers and spleens of immunized mice were fixed with 4% paraformaldehyde (Solarbio, Beijing, China), and then paraffin-sectioned and stained using the Hematoxylin-Eosin (HE) Staining Kit (Solarbio) in accordance with the manufacturer’s instructions. Briefly, the fixed samples were embedded and sectioned, then conventional dewaxing and hydration were performed. The tissue sections were stained with HE, after which, dehydration, cleaning for transparency, and neutral resin sealing were performed successively.

Reviewer 2 Report

The manuscript by Huang et al. 2020 describes a  method to generate vaccine of brucella abortus using O-linked glycosylation system in Yersinia enterocolitica.

Although the system used in this study is not new, but the target bacteria is. The use of less pathogenic bacteria is indeed a smart way to avoid the large-scale culturing of highly pathogenic one. The manuscript provides interesting result, and the result and discussion has sufficient data to support their hypothesis. Therefore, I recommend this MS to be accepted in Microorganism journal.

However, the MS contains some typos and incorrect english sentence for example:

Line 89: Serious should be series

Line 177: This safer strain sound awkward, please change it.

Line 339: failed to produce

Line 375: urgently? Do you mean urgently needed?

Overall, I recommend the MS to be further checked by a native English speaker prior to publication.

Additional comment,

Figure 1A. Why the glycosylated one showed very thick band? Did you adjust the to the same concentration of CTB? If so, please describes in the methodology.

Author Response

Thank you for your comments, we have uploaded the point-by-point response as a Word file.
